# COVID-19 Vaccine Effectiveness at a Referral Hospital in Northern Peru: A Retrospective Cohort Study

**DOI:** 10.3390/vaccines10050812

**Published:** 2022-05-20

**Authors:** Mario J. Valladares-Garrido, Sandra Zeña-Ñañez, C. Ichiro Peralta, Jacqueline B. Puicón-Suárez, Cristian Díaz-Vélez, Virgilio E. Failoc-Rojas

**Affiliations:** 1Vicerrectorado de Investigación, Universidad Norbert Wiener, Lima 15046, Peru; mario.valladares@uwiener.edu.pe; 2Hospital Regional Lambayeque, Chiclayo 14012, Peru; 3Instituto de Evaluación de Tecnologías en Salud e Investigación-IETSI, EsSalud, Lima 15072, Peru; cdiazv3@upao.edu.pe; 4School of Medicine, Universidad Continental, Huancayo 12001, Peru; szena@continental.edu.pe; 5School of Medicine, Universidad Nacional Federico Villarreal, Lima 15088, Peru; 2015027078@unfv.edu.pe; 6Sociedad Científica de Estudiantes de Medicina, Universidad Nacional Pedro Ruiz Gallo, Lambayeque 14013, Peru; jpuicons@unprg.edu.pe; 7School of Medicine, Universidad Privada Antenor Orrego, Trujillo 13008, Peru; 8Vicerrectorado de Investigación, Universidad San Ignacio de Loyola, Lima 15024, Peru

**Keywords:** vaccination, COVID-19, mortality, vaccine effectiveness, Peru

## Abstract

COVID-19 vaccines have achieved a significant reduction in mortality, yet objective estimates are needed in specific settings. We aimed to determine the effectiveness of COVID-19 vaccination at a referral hospital in Lambayeque, Peru. We conducted a retrospective cohort study from February to September 2021. We included hospitalized patients with COVID-19, whose data were stored in NotiWeb, a patient data system of the Peruvian Ministry of Health. We applied a propensity score-weighting method according to baseline characteristics of patients, and estimated hazard ratios (HR) using Cox regression models. Of 1553 participants, the average age was 55 years (SD: 16.8), 907 (58%) were male, and 592 (38%) deceased at 28-day follow-up. Before hospital admission, 74 (4.8%) had been immunized with at least one vaccine dose. Effectiveness against death in vaccinated patients was 50% at 90-day follow-up (weighted HR 0.50, 95% CI 0.28–0.89). Our results support the effectiveness of COVID-19 vaccination against death and provide information after early immunization in Peru.

## 1. Introduction

By September 2021, the coronavirus-19 (COVID-19) pandemic has caused 4.7 million deaths all around the world [1] and 2,092,275 in America [2]. In the same period, Peru was the sixth country with the highest number of deaths due to COVID-19 in the world, with 199,314 deaths [3]. SARS-CoV-2 is the virus that causes COVID-19; this is a betacoronavirus that has a spike (S) structural glycoprotein distributed throughout its surface. This protein gives the virus the capacity to bind to the mucous membrane through the angiotensin-converting enzyme 2 (ACE2) receptor [4,5] in the respiratory tract. This process generates a local inflammatory response which can progress to a systemic disease [5].

With current information, it has been possible to conduct studies for the creation of vaccines against COVID-19, aiming to generate an artificial active immune response in patients [6]. Vaccination is one of the most effective prevention strategies in public health since it significantly reduces the incidence of multiple communicable diseases [7]. In Peru, Pfizer BioNTech, Sinopharm, and Oxford AstraZeneca vaccines have been acquired. They are based on mRNA, inactivated virus, and viral vector (non-replicating), respectively [8]. Pfizer vaccine has demonstrated 95% protection against COVID-19 in people aged 16 or over, after two shots, during an interval of 21 days [9]. On the other hand, Sinopharm vaccine has proved to have 79% efficacy after the administration of a second shot after an interval of 21 days [10]. AstraZeneca vaccine has 76% efficacy against symptomatic SARS-CoV-2 infection, after the administration of a second shot in an interval of 4–12 weeks [11].

In general, there is little research on the effectiveness of COVID-19 vaccines assessing different clinical outcomes in hospitalized patients. One study [12] showed effectiveness of mRNA vaccines of 88%, 90%, and 91% (without identifying viral variants) regarding hospitalization, UCI admission, and urgent care visit, respectively. However, clinical variables were not considered for the analysis [12]. Studies have demonstrated that vaccinated people present immunity up to 6 months after a second dose, in comparison with unvaccinated people [13]. However, with the emergence of SARS-CoV-2 variants, vaccine effectiveness has decreased after some months of second dose administration. For example, Israel have shown a high frequency of positive cases (98%) of the Delta variant among vaccinated individuals [14].

Due to the limited information in the national and international context, we aimed to evaluate the effectiveness of COVID-19 vaccines in hospitalized patients from a referral health facility in northern Peru. Additionally, we analyzed clinical variables related to the disease.

## 2. Materials and Methods

We conducted a retrospective cohort study based on the analysis of secondary data. The data corresponded to patients hospitalized between February and September 2021 in the Hospital Regional Lambayeque, in northern Peru. The primary study aimed to evaluate the prognostic factors for death in patients who sought treatment at the hospital during the health emergency.

The population consisted of patients hospitalized for suspected COVID-19 at the Hospital Regional Lambayeque from 1 February to 30 September 2021. This period coincided with the emergence of the second epidemic wave in Peru, in which the gamma variant predominated. The sample consisted of patients with a COVID-19 diagnosis, confirmed through a molecular (RT-PCR) or an antigen test. SARS-CoV-2 infection was confirmed through the record of the molecular or antigen test in the epidemiological surveillance systems of the Peruvian Center for Epidemiology, Prevention and Disease Control (CDC Peru), National Network of Public Health Laboratory System (NetLab2), and the Integrated System of COVID-19 (SISCOVID). Patients aged 18 years or older were included. Records with incomplete data regarding the variables of interest were excluded.

We exported data from electronic records of the Epidemiological Office at the Hospital Regional de Lambayeque. These files consisted of epidemiological notification data of COVID-19 (clinical–epidemiological research records and COVID-19 hospitalization and deaths surveillance records), which have been stored in the epidemiological surveillance systems of CDC Peru. Additionally, we used physical records to perform quality control and to resolve inconsistent, out-of-range, and incomplete data identified in the electronic files.

We used clinical–epidemiological research records that included variables related to epidemiological antecedents, clinical variables (signs, symptoms, and pathological personal antecedents), vaccination status, hospitalization variables (hospitalization date, date of discharge, and date of death), and evolution (recovered, death).

The outcome variable was COVID-19 mortality, which was defined as patients who died from COVID-19 as recorded in the clinical–epidemiological research files of the Epidemiological Surveillance System (NotiWeb-COVID) of CDC Peru, for at least the virological or serological criterion. COVID-19 deaths were considered confirmed if they complied with at least one criterion: virological (molecular test or antigen reactive to SARS-CoV-2), serological (positive serological IgM or IgM/IgG test for SARS-CoV-2), radiological (radiological, tomographic, or nuclear magnetic resonance imaging compatible with COVID-19 pneumonia), epidemiological (link with a confirmed case of COVID-19), or clinical (clinical picture compatible with COVID-19 disease). Death was also confirmed if it was recorded in the National Death System of Peru (SINADEF). Death confirmed by SINADEF (https://www.minsa.gob.pe/defunciones/ accessed on 26 September 2020) was corroborated using any of the following ICD-10 codes: U071, U072, B342, B972, or the search terms “coronavirus”, “COV-2”, “COV2”, “COVID”, and “SARS”.

The exposure variable was vaccination against COVID-19, defined as having received an unspecified COVID-19 vaccine (Pfizer, Sinopharm, or AstraZeneca), recorded in the vaccination status database of the Peruvian Ministry of Health (MINSA). An individual was considered unvaccinated if there was no evidence of any shot in their vaccination card. Patients partially vaccinated against COVID-19 were those who had received only the first dose. Finally, patients considered as immunized were those who had received two doses of the COVID-19 vaccine and it was recorded in their vaccination cards.

Secondary covariates were age (years), sex (male, female), clinical picture at the onset of the disease (dyspnea, fever, cough), and presence of comorbidities (arterial hypertension, type 2 diabetes, chronic kidney disease, cancer, and neurological disorder).

Statistical analysis was performed in Stata v.17.0 (StataCorp LP, College Station, TX, USA). In the descriptive analysis, frequencies and percentages were shown for categorical variables. For numerical variables, the best measures of central tendency and dispersion were reported.

Bivariate analysis was performed to evaluate the association between potential clinical–epidemiological factors and COVID-19 mortality after 28 days and 90 days of hospitalization. The chi-square test was used for categorical variables, after the evaluation of the expected frequency assumption. In addition, the Student’s *t*-test for independent samples was used for numerical variables, after evaluation of normal distribution. We set a significance level of 5%.

To simulate random assignment in clinical trials, we applied a propensity score weighting to multivalent treatments using a machine learning approach that allowed us to calculate balanced differences for each control and treatment group according to reference covariates [15]. The propensity score (PS) function estimates the subject’s probability of receiving a given treatment (vaccines) based on their pre-treatment characteristics.

The obtained PSs were transformed into propensity score weighting (PSW) to estimate the mean allocation effect on treated subjects [16]. This method sets the PSW at 1 for each participant in the treatment group and calculates the PSW for the control group using the probabilities in the reference group (PSW = propensity score/(propensity score-1)).

Cox proportional hazards regression models were constructed to estimate weighted and unweighted hazard ratios (HR) by PSW and 95% CI, with the purpose of identifying prognostic factors associated with mortality. A Kaplan–Meier survival curve was constructed using the log-rank-test. Time until the event of interest (death) was defined as the time from hospital admission to notification of date of death after 28 days and 90 days of hospitalization. Mortality data at 90 days included data for deaths after 28 days (i.e., death registration was cumulative).

This study was approved by the Ethics Committee of the Hospital Regional Lambayeque. Anonymized codes were used to preserve the confidentiality of personal data of patients selected for the study. We followed the ethical principles of the Declaration of Helsinki. Additionally, the primary study has been registered in the repository of the Peruvian National Institute of Health (PRISA).

## 3. Results

We enrolled 1553 participants who were hospitalized from February to September 2021, in a referral hospital in Lambayeque, Peru.

Of the total, only 74 (4.8%) were immunized with at least one dose of the vaccines at the time of hospitalization. The average age was 55.2 (±16.8) years, with a proportion of males to females of 3:2. The most frequent symptom was cough (73.5%) and the most frequent comorbidity was hypertension (6.3%). Of those hospitalized, 592 (38.1%) died after 28 days and 614 (39.5%) after 90 days. Details are presented in Table 1.

Of those vaccinated, only 16 (21.6%) and 19 (25.7%) died 28 days and 90 days hospitalization respectively, which was lower than the frequency of those unvaccinated. Those who died after 28 days or 90 days had an average age 13 years older than the average age of those who did not die. More than half of those with dyspnea at the onset of the disease died after 28 days or 90 days (*p*-value < 0.001). Eight out of ten hospitalized patients with hypertension, type 2 diabetes, chronic kidney disease, or a neurological disorder died after 28 days or 90 days. This association was statistically significant (*p*-value < 0.001). More information is provided in Table 2.

The hazard of death was 49% and 42% lower in those who had at least one dose after 28 days and 90 days of hospitalization, respectively, compared with those who were unvaccinated, and was significant after adjusting for sex, age, symptoms at onset of disease (dyspnea, fever, and cough), and comorbidities (hypertension, type 2 diabetes, chronic kidney disease, and cancer) (aHR = 58%; 95% CI: 16–69% and aHR = 53%; 95% CI: 15–60%, respectively). When there is an age increase of 1 year, the adjusted hazard of death after 28 days and 90 days increases by 3% and 4% respectively (*p*-value < 0.001). The presence of hypertension and diabetes were risk factors for death after 28 days (HR = 3.74; 95% IC: 2.95–4.75) and 90 days (HR = 3.68; 95% CI: 2.90–4.67). This association was still significant in the adjusted model (aHR = 2.18; 95% IC: 1.70–2.81 and aHR = 2.14; 95% IC: 1.67–2.76). See Table 3.

After using a weighting model by vaccination exposure factors, the HR after 28 days was 0.49 (95% CI 0.26–0.89) and the HR after 90 days was 0.50 (95% CI 0.28–0.89). See Table 4 for more information. Additional information on propensity score weighting can be found in Appendix A.

## 4. Discussion

We observed that percentage of individuals vaccinated against COVID-19 was low (4.8%). This could be because the study was conducted at the beginning of the vaccination period in Peru, but also because the patients may have shown a possible lack of knowledge or refusal of immunization. The most used vaccine was Pfizer’s, and this is consistent with the average age of participants (55.2 years), since the country prioritized Pfizer vaccines in this population during the first campaigns. However, 4 out of 10 participants died at the end of follow-up, which could be partly related to the second wave of COVID-19 in Peru. This suggests that there were a considerable number of deaths among unvaccinated individuals, which is more evident in older people and those with comorbidities.

We found that vaccines were estimated to be 50% effective against death at 90-day follow-up. This represents a lower risk of death in those who were vaccinated with at least one dose. Studies in the United Kingdom, United States, Israel, and Canada reported 64–100% effectiveness against death for one dose, and between 94% and 100% for two doses [17]. These studies evaluated the effectiveness of Pfizer, AstraZeneca, and Moderna vaccines, mostly against the alpha variant. However, there is little evidence on the effectiveness of the other types of vaccines and against the variants of concern in relation to mortality. This could affect countries like Peru, which has a diverse genomic distribution of the virus.

Most effectiveness studies showed that the risk of death is even lower in patients vaccinated with two doses. Our results suggested that patients with one dose may have a similar risk of death as those who received two doses. Even if one dose does not achieve the immune protection needed to protect against infection, the risk of death may be significantly reduced. This situation depends on other factors, such as previous exposure to the virus, immune status, or presence of comorbidities. In addition, time of immune protection may vary because of type of vaccine and individual biological factors. Our results are limited due to a short follow-up time. However, administration of one dose could at least prevent collapse of health systems in the event of COVID-19 waves.

Our study showed that the lower risk of death among vaccinated individuals was maintained at 90-day follow-up. This result is comparable with studies using Pfizer’s vaccine, which showed an adequate level of immune protection after 5 months [14]. However, the reported effectiveness was only against delta variant. The present study focused on the gamma variant, which predominated in the northern region of Peru. Evasion of immunity increases with mutations of the viral antigenic regions, explaining the reduced effectiveness of the vaccines. A relationship of this reduction with increase in deaths needs to be estimated over longer follow-up periods. However, application of a booster dose has proven to be an effective measure to mitigate the effect of variants of concern [18,19].

Figure 1 shows that the probability of survival in unvaccinated individuals drops significantly in the first 20 days of follow-up. In contrast, the probability of survival in vaccinated individuals decreases insidiously over a longer time span. This suggests that vaccination confers rapid immune protection that may prevent death in the first days of disease. Moreover, effectiveness against mortality is maintained from day 40 to day 90 of follow-up, indicating a relatively prolonged period of immunity.

Comorbidities play an important role in immune mediation against COVID-19. The present study demonstrated that diseases—such as hypertension, diabetes, or cancer—increase the risk of death up to three times. An increased severity and risk of death by COVID-19 in unvaccinated individuals with diabetes has been previously reported [20,21]. Likewise, vaccinated individuals with diabetes, hypertension, or obesity showed similar levels of IgG and neutralizing antibodies as those without comorbidities [22]. However, antibodies may decline rapidly in this population due to immune impairment caused by a dysfunctional physiological state [23,24]. Consequently, vaccine administration should be prioritized in individuals with comorbidities. A different number of doses and time between each dose could also be indicated in this group.

Immune status may be altered by susceptibility to other viruses. It has been observed that those vaccinated against influenza and pneumococcus had indirect benefits with respect to the risk of COVID-19 [25]. In addition, co-infection with SARS-CoV-2 and influenza has been shown to have no effect on mortality [26]. This suggests that vaccination against different respiratory pathogens may confer an additional immune protection against COVID-19. Despite the limited evidence, vaccination against other pathogens could help decrease the high demand for hospitalization, which has been a critical situation in Peru during the pandemic. Furthermore, mitigation of influenza and pneumonia cases would allow differentiation of specific cases of COVID-19. Further evidence is needed to determine the role of other vaccines against COVID-19, as well as co-infection scenarios.

The current vaccination situation is favorable for preventing deaths from COVID-19. It is important to recognize the effect that vaccines are having outside the clinical trial contexts. This study demonstrates a significant reduction in deaths in a specific population in the northern region of Peru, where the gamma variant predominated. This complements the favorable results shown in other countries. However, considering that almost all patients in this study received mRNA vaccines, there is uncertainty about the effectiveness of other types of vaccines. This is particularly relevant because numerous immunizations in Peru have also included inactivated virus formulations.

The study has several limitations. First, the maximum follow-up period was only 90 days, and there could be differences in the number of deaths in a longer period. Second, the risk of death could not be evaluated according to the number of doses and type of vaccine due to the small number of vaccinated participants (for details on Kaplan-Meier survival curves by vaccine type, see Appendix A). Third, hazard ratios may be influenced by unmeasured community- or hospital-level variables such as self-medication, delayed hospitalization, smoking, cancer treatment, and objective clinical data (e.g., SpO2 and body temperature). However, we included important clinical variables that strengthen the validity of the results. In addition, this study included a large number of participants who came from a referral hospital in northern Peru.

## 5. Conclusions

There was a low percentage of hospitalized patients who received a COVID-19 vaccine during the first months of immunization in Peru. Even though mortality was considerably high, risk of death was significantly lower in vaccinated patients in comparison to those unvaccinated. The vaccine effectiveness against death was maintained at 90 days of follow-up. These results highlight the importance of vaccination in a region where the pandemic has had a significant impact. Future studies could include more specific information on immune status, such as IgG titer and neutralizing antibodies. It would also be useful to evaluate vaccine effectiveness with outcomes such as severity of disease or prevention of infection. Moreover, analyses in particular age groups and with specific vaccines are required. Inclusion of schemes with a combination of different types of vaccines raises the need for new evidence of their effectiveness.

## Figures and Tables

**Figure 1 vaccines-10-00812-f001:**
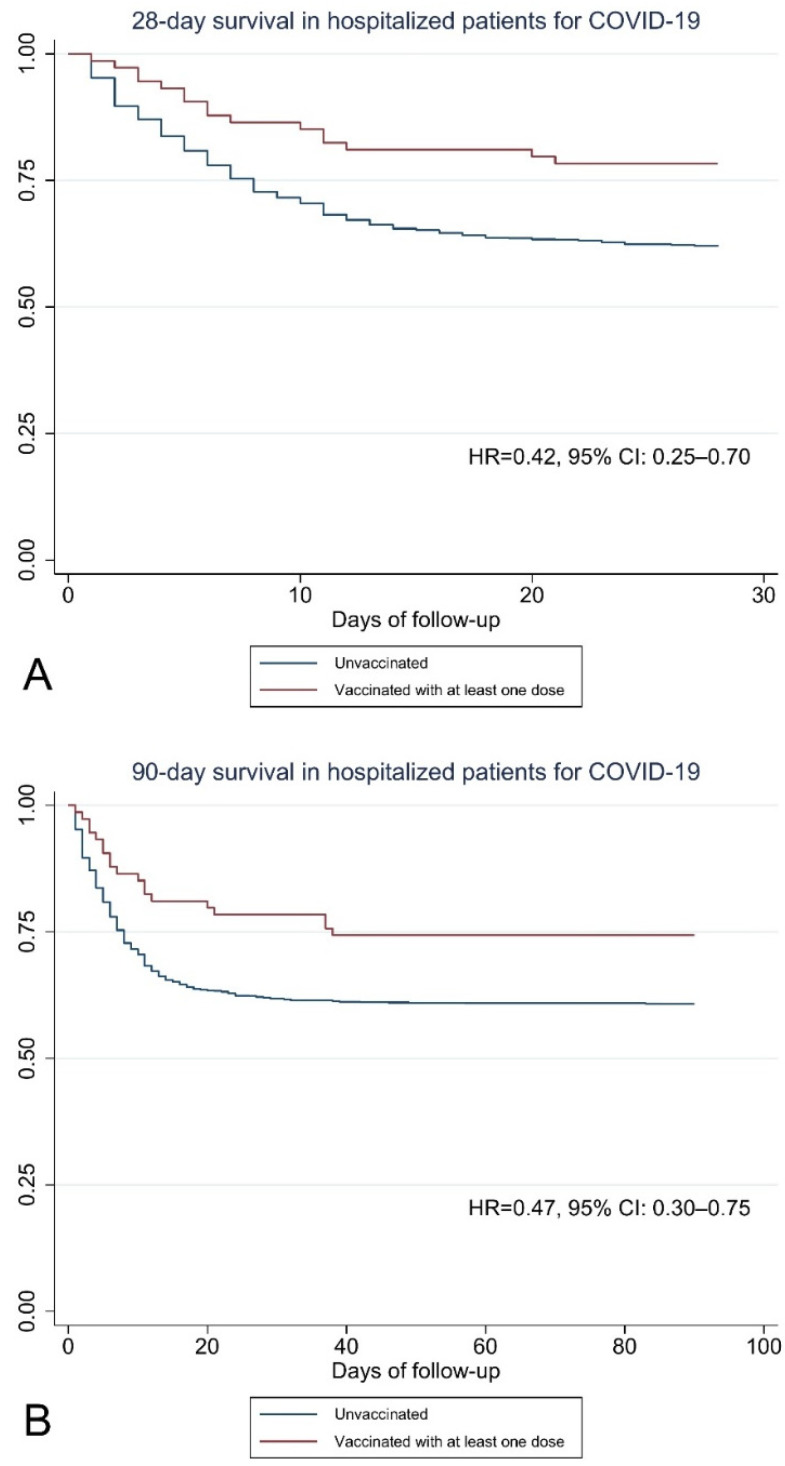
Survival of patients hospitalized for COVID-19 according to vaccination status. (**A**) Survival after 28 days. (**B**) Survival after 90 days.

**Table 1 vaccines-10-00812-t001:** Clinical and epidemiological characteristics of hospitalized COVID-19 patients (*n* = 1553).

Characteristics			*n* (%)
Vaccinated			
	Unvaccinated		1480 (95.2)
	Only the first dose		43 (2.8)
	First and second dose		31 (2.0)
Vaccine type			
	Pfizer		64 (86.5)
	Sinopharm		8 (10.8)
	AstraZeneca		2 (2.7)
Age (years) *			55.2 ± 16.8
Sex			
	Female		647 (41.6)
	Male		907 (58.4)
Medical insurance			
	SIS		899 (57.8)
	EsSalud		152 (9.8)
	Others		26 (1.7)
	No medical insurance		477 (30.7)
Symptoms at the onset of the disease			
	Dyspnea		
		No	706 (45.5)
		Yes	847 (54.5)
	Fever		
		No	989 (63.7)
		Yes	564 (36.3)
	Cough		
		No	411 (26.5)
		Yes	1142 (73.5)
Comorbidity			
	Hypertension **		
		No	1454 (93.7)
		Yes	97 (6.3)
	Type 2 diabetes **		
		No	1521 (98.1)
		Yes	30 (1.9)
	Chronic kidney disease **		
		No	1533 (98.8)
		Yes	18 (1.2)
	Obesity **		
		No	1516 (97.7)
		Yes	35 (2.3)
	Cancer **		
		No	1516 (97.7)
		Breast	10 (0.65)
		Cervical	3 (0.19)
		Prostate	3 (0.19)
		Stomach	2 (0.13)
		Brain	2 (0.13)
		Non-Hodgkin’s lymphoma	2 (0.13)
		Unspecified or uncommon site	13 (0.83)
	Neurological *		
		No	1526 (98.4)
		Yes	25 (1.6)
Death after 28 days			592 (38.1)
Death after 90 days			614 (39.5)

* Mean ± standard deviation. ** Some variables can have missing data. SIS: Seguro Integral de Salud.

**Table 2 vaccines-10-00812-t002:** Association between vaccination against COVID-19 and mortality in bivariate analysis.

Variables			Mortality after 28 Days			Mortality after 90 Days		
			No (n = 961)	Yes (*n* = 592)	*p*	No (n = 939)	Yes (*n* = 614)	*p*
			*n* (%)	*n* (%)		*n* (%)	*n* (%)	
Vaccinated *					0.003			0.012
	Unvaccinated		903 (61.1)	576 (38.9)		884 (59.8)	595 (40.2)	
	At least one dose		58 (78.4)	16 (21.6)		55 (74.3)	19 (25.7)	
Age (years) **			50.4 ± 16.2	63.1 ± 14.7	0.001	50.1 ± 16.2	63.1 ± 14.6	0.001
Sex *					0.001			0.001
	Female		447 (69.1)	200 (30.9)		437 (67.5)	210 (32.5)	
	Male		514 (56.7)	392 (43.3)		502 (55.4)	404 (44.6)	
Symptoms at the onset of the disease								
	Dyspnea *				0.001			0.001
		No	564 (78.9)	142 (20.1)		556 (78.7)	150 (21.3)	
		Yes	397 (46.9)	450 (53.1)		383 (45.2)	464 (54.8)	
	Fever *				0.009			0.013
		No	636 (64.3)	353 (35.7)		621 (62.8)	368 (37.2)	
		Yes	325 (57.6)	239 (42.4)		318 (56.4)	246 (43.6)	
	Cough *				0.001			0.001
		No	198 (48.1)	213 (51.8)		190 (46.2)	221 (53.8)	
		Yes	763 (66.8)	379 (33.2)		749 (65.6)	393 (34.4)	
Comorbidity								
	Hypertension *				0.001			0.001
		No	945 (65.0)	509 (35.0)		923 (63.5)	531 (36.5)	
		Yes	14 (14.4)	83 (85.6)		14 (14.4)	83 (85.6)	
	Type 2 diabetes *				0.001			0.001
		No	953 (62.7)	568 (37.3)		931 (61.2)	590 (38.8)	
		Yes	6 (20.0)	24 (80.0)		6 (20.0)	24 (80.0)	
	Chronic kidney disease *				0.001			<0.001
		No	955 (62.3)	578 (37.7)		934 (60.9)	599 (39.1)	
		Yes	4 (22.2)	14 (77.8)		3 (16.7)	15 (83.3)	
	Obesity *				<0.001			<0.001
		No	954 (62.9)	562 (37.1)		932 (61.5)	584 (38.5)	
		Yes	5 (14.3)	30 (85.7)		5 (14.3)	30 (85.7)	
	Cancer *				0.822			0.453
		No	938 (61.9)	578 (38.1)		918 (60.5)	598 (39.5)	
		Yes	21 (60.0)	14 (40.0)		19 (54.3)	16 (45.7)	
	Neurological *				<0.001			<0.001
		No	956 (62.6)	570 (37.4)		934 (61.2)	592 (39.8)	
		Yes	3 (12.0)	22 (88.0)		3 (12.0)	22 (88.0)	

* *p*-value obtained through chi-squared test. ** *p*-value obtained through Student’s *t*-test for independent samples and equal two-tailed variances.

**Table 3 vaccines-10-00812-t003:** Association between COVID-19 vaccination and mortality in multivariate analysis.

		After 28 Days				After 90 Days			
Characteristics		HR	95% CI	aHR *	95% CI	HR	95% CI	aHR *	95% CI
Vaccinated									
	Unvaccinated	Ref.		Ref.		Ref.		Ref.	
	At least one dose	0.51	0.31–0.84	0.42	0.25–0.70	0.58	0.37–0.92	0.47	0.30–0.75
Age (years)		1.03	1.03–1.04	1.03	1.03–1.04	1.04	1.03–1.04	1.03	1.03–1.04
Sex									
	Female	Ref.				Ref.			
	Male	1.49	1.25–1.78			1.47	1.24–1.74		
Symptoms at the onset of the disease									
	Dyspnea (yes)	3.38	2.79–4.10			3.33	2.76–4.03		
	Fever (yes)	1.26	1.07–1.49			1.24	1.06–1.47		
	Cough (yes)	0.56	0.47–0.66			0.56	0.47–0.66		
Comorbidity									
	Hypertension (yes)	3.74	2.95–4.75			3.68	2.90–4.67		
	Type 2 diabetes (yes)	3.03	1.98–4.64			2.97	1.94–4.55		
	Chronic kidney disease (yes)	2.51	1.47–4.26			2.65	1.59–4.42		
	Obesity (yes)	3.80	2.61–5.53			3.74	2.57–5.44		
	Cancer (yes)	1.06	0.62–1.80			1.17	0.71–1.92		
	Neurological (yes)	3.71	2.42–5.69			3.66	2.39–5.61		

* Adjusted for age, sex, dyspnea, fever, cough, hypertension, type 2 diabetes, chronic kidney disease, obesity, cancer, and neurological disorder. HR: hazard ratio; aHR: adjusted hazard ratio; 95% CI: 95% confidence interval.

**Table 4 vaccines-10-00812-t004:** COVID-19 vaccine effectiveness against death and weighted hazard ratios for mortality.

Characteristics		After 28 Days				After 90 Days			
		HR *	95% CI	*p*	Effectiveness	HR *	95% CI	*p*	Effectiveness
Vaccinated									
	Unweighted	0.42	0.25–0.70	<0.001	58.0%	0.47	0.30–0.75	0.002	53.0%
	Weighted	0.49	0.26–0.89	0.019	51.0%	0.50	0.28–0.89	0.019	50.0%

* Adjusted for age, sex, dyspnea, fever, cough, hypertension, type 2 diabetes, chronic kidney disease, obesity, cancer, and neurological disorder. HR: hazard ratio. 95% CI: 95% confidence interval.

## Data Availability

The dataset generated and analyzed during the current study is not publicly available because the ethics committee has not provided permission/authorization to publicly share the data but are available from the corresponding author on reasonable request.

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
