# Peer review of "COVID-19 Vaccine Effectiveness at a Referral Hospital in Northern Peru: A Retrospective Cohort Study"

_vaccines, 2022, doi:10.3390/vaccines10050812_

Round 1

Reviewer 1 Report

Estimated Authors,

I've read your paper with great interest. Your report on the mortality from COVID-19 in Peru, Region of Lambayeque, during 2021, clearly stresses how vaccination do starkly reduce mortality in cases of SARS-CoV-2 infection (aHR 0.42, 95%CI 0.25-0.70 at 28 days, aHR 0.47, 95%CI 0.30-0.75 at 90 days), no question about it.

From my point of view, this paper deserves a prompt publication from Vaccines, as it reports on a settings (i.e. a country with a background high mortality from SARS-CoV-2) that may profit at most from implementation of more equality-based vaccination strategies.

However, some improvements are, from my point of view, required. 

First of all, the most significant one:

As shown in Table 3, you've calculated HR both as a crude estimate and as an adjusted one. aHR was adjusted by sex age, but also hypertension, diabestes, chronic renal disease and Cancer. But table 3 reports also on aHR FOR hypertension, obesity, cancer. In other words, your HR was rather a multivariable one than an adjusted one. Moreover, why Table 3 did not include estimates for obesity and neurological diseases? Some clarifications are required.

Similarly, please clarify that mortality at 90 days did include also deaths after 28 days, as the main text may cause come misunderstandings.

Reviewer 2 Report

This study retrospectively examines the impact of COVID-19 vaccination status on mortality in Peru. Currently, there is a trend to make vaccination a prerequisite for the resumption of economic activity; however, it is necessary to collect a wide range of information on the different types of vaccines administered in each country. In this sense, the study should be evaluated.

As the survey period of this study was from February to September 2021, the main target seemed to be the δ strain.

Table 1 shows the epidemiological characteristics of the subjects. For males, a history of smoking is a possible confounding factor, so information on smoking history should be added if available.

Dyspnea and fever are listed as physical findings, but objective data such as SpO2 and body temperature at the time of the admission should be included if available.

In addition, the presence or absence of cancer is mentioned in the current medical history, but which site of cancer and whether the patient has been treated with anticancer drugs should be stated.

In Fig. 1, the "Vaccinated" section states that "The most used vaccine was Pfizer's," but Table 1 lists 64 cases by Pfizer, 8 cases by AstraZeneca, and 2 cases by Sinopharm. If the specific numbers are known, they should not be lumped together as "vaccinated, and it is desirable to indicate each type of case separately.

Round 2

Reviewer 2 Report

The authors have made appropriate additions and corrections based on the points raised. In particular, the contents of the supplementary files will be of great interest to clinicians worldwide.